# Impact of Maternal Pre-Pregnancy Underweight on Cord Blood Metabolome: An Analysis of the Population-Based Survey of Neonates in Pomerania (SNiP)

**DOI:** 10.3390/ijms25147552

**Published:** 2024-07-10

**Authors:** Alexander Lichtwald, Till Ittermann, Nele Friedrich, Anja Erika Lange, Theresa Winter, Claudia Kolbe, Heike Allenberg, Matthias Nauck, Matthias Heckmann

**Affiliations:** 1Department of Neonatology and Pediatric Intensive Care, University Medicine Greifswald, 17475 Greifswald, Germany; anja.lange@med.uni-greifswald.de (A.E.L.); heike.allenberg@med.uni-greifswald.de (H.A.); 2Institute for Community Medicine, Division SHIP—Clinical Epidemiological Research, University Medicine Greifswald, 17475 Greifswald, Germany; till.ittermann@med.uni-greifswald.de; 3Institute for Clinical Chemistry and Laboratory Medicine, University Medicine Greifswald, 17475 Greifswald, Germany; nele.friedrich@med.uni-greifswald.de (N.F.); theresa.winter@med.uni-greifswald.de (T.W.); matthias.nauck@med.uni-greifswald.de (M.N.); 4German Centre for Cardiovascular Research (DZHK), Partner Site Greifswald, 17475 Greifswald, Germany; 5Department of Gynecology and Obstetrics, University Medicine Greifswald, 17475 Greifswald, Germany; claudia.kolbe@med.uni-greifswald.de; 6German Centre for Child and Adolescent Health (DZKL), Partner Site Greifswald/Rostock, 17475 Greifswald, Germany

**Keywords:** pre-pregnancy BMI, gestational weight gain, placental ratio, neonatal outcome, metabolome, lipoprotein, amino acids

## Abstract

Intrauterine growth restriction leads to an altered lipid and amino acid profile in the cord blood at the end of pregnancy. Pre-pregnancy underweight is an early risk factor for impaired fetal growth. The aim of this study was to investigate whether a pre-pregnancy body mass index (ppBMI) of <18.5 kg/m^2^, as early as at the beginning of pregnancy, is associated with changes in the umbilical cord metabolome. In a sample of the Survey of Neonates in Pomerania (SNIP) birth cohort, the cord blood metabolome of n = 240 newborns of mothers with a ppBMI of <18.5 kg/m^2^ with n = 208 controls (ppBMI of 18.5–24.9 kg/m^2^) was measured by NMR spectrometry. A maternal ppBMI of <18.5 kg/m^2^ was associated with increased concentrations of HDL4 cholesterol, HDL4 phospholipids, VLDL5 cholesterol, HDL 2, and HDL4 Apo-A1, as well as decreased VLDL triglycerides and HDL2 free cholesterol. A ppBMI of <18.5 kg/m^2^ combined with poor intrauterine growth (a gestational weight gain (GWG) < 25th percentile) was associated with decreased concentrations of total cholesterol; cholesterol transporting lipoproteins (LDL4, LDL6, LDL free cholesterol, and HDL2 free cholesterol); LDL4 Apo-B; total Apo-A2; and HDL3 Apo-A2. In conclusion, maternal underweight at the beginning of pregnancy already results in metabolic changes in the lipid profile in the cord blood, but the pattern changes when poor GWG is followed by pre-pregnancy underweight.

## 1. Introduction

Translational research has shown strong associations between fetal growth and development and the fetal programming of health and diseases in later life [1]. The periconceptional period, which is defined as the time window of 14 weeks before the 10 weeks after conception, is particularly important. This period covers the vulnerable processes of gametogenesis, embryogenesis, and the initiation of placentation, thus representing a critical time window for exposure with potentially large effects during the entire prenatal, as well as postnatal, life course. In this context, a large meta-analysis, using the data from 196,670 participants within 25 cohort studies in Europe and North America, showed that maternal pre-pregnancy BMI (ppBMI) was more strongly associated with adverse maternal and infant short-term outcomes than gestational weight gain (GWG) [2]. Furthermore, a higher maternal ppBMI and GWG were associated with an increased risk of childhood overweight/obesity, with the strongest effects occurring at later ages [3]. Here, the additional effect of gestational weight gain in women who are overweight or obese before pregnancy was also small. However, the children of underweight mothers, as well as for overweight and obese mothers, are also at risk of adverse outcomes [4,5,6,7,8,9]. The prevalence of maternal underweight in industrialized countries is lower compared to maternal overweight, which can range from 4% to 12% depending on the reported classification of being underweight [2,9,10,11]. In the Survey of Neonates in Pomerania (SNiP) birth cohort study, maternal underweight was associated with a higher odds ratios for a low birthweight (<2500 g, LBW), late preterm birth, neonatal asphyxia, admission to neonatal care, and a lower placental weight [12]. Placental weight is an important determinant for fetal growth and thus for birth weight. An impaired placental function leads to disrupted fetal growth due to impaired uteroplacental and umbilical blood flow in intrauterine growth restriction (IUGR), thus leading to altered placental function, which includes the transport of critical nutrients from the mother to the fetus [13,14]. Furthermore, the effects of fetal undernutrition depends on its timing during gestation and the organs and tissues undergoing critical periods of development at that time. Early gestation appeared to be the most vulnerable [15]. Another well-established decisive factor of fetal growth is GWG in relation to maternal pre-pregnancy weight. An inadequate GWG predisposes one for IUGR and a lower birth weight [10,16]. Children born with low birthweight also have a higher risk of non-communicable diseases such as cardiovascular diseases, diabetes mellitus, and metabolic syndrome in later life [17,18,19,20,21]. Associations between small placentas and risks of coronary heart disease and diabetes mellitus have also been found [22].

The intrauterine environment and its relation to maternal phenotypes is reflected by the fetal metabolome. The fetal metabolome comprises the totality of all metabolites at a specific point in time. Analysis of the fetal metabolome provides the opportunity to reconcile fetal gene expression, epigenetic changes, and environmental influences with the metabolic phenotype of the fetus. New insights into the fetal metabolome, therefore, enable a better understanding of physiological and pathological processes. The fetal metabolome is generated primarily from two sources: the transplacental transfer of metabolites and fetal synthesis, and the metabolism of circulating metabolites. Altered lipid and lipoprotein profiles have been reported in neonates after IUGR and those that are born small for gestational age (SGA) in cord blood or during the first days of life. Miranda et al. [23] observed higher plasma concentrations of cholesterol and triglycerides transporting lipoproteins, as well as an increased number of very-low density lipoprotein (VLDL) particles in intrauterine growth-restricted fetuses. Wang et al. [24] reported higher concentrations of triglycerides, total cholesterol (TC), and low-density lipoprotein cholesterol (LDL), but not high-density lipoprotein cholesterol (HDL-C) in SGA newborns in comparison to the appropriate-for-gestational-age (AGA) newborns at 72 h of life. In contrast, Pecks et al. found lower fetal HDL-C and TC concentrations in IUGR infants compared to AGA or SGA infants without IUGR [25]. Furthermore, a significant reduction in amino-acid fetal–maternal gradients and in umbilical veno-arterial differences has been demonstrated in IUGR pregnancies [26]. Small-for-gestational-age fetuses have significantly lower concentrations of the branched chain amino acids valine, leucine, and isoleucine, as well as of lysine and serine [27]. Furthermore, tyrosine, phenylalanine, glutamate, and glutamine have been reported to be altered in the metabolome of SGA and IUGR infants depending on the severity of IUGR and the timing of blood sampling [28,29].

In summary, intrauterine growth restriction leads to an altered lipid and amino acid profile in the cord blood at the end of pregnancy. Pre-pregnancy underweight is an early risk factor for impaired fetal growth and later adverse outcomes. But studies of the impact of maternal BMI on the fetal metabolome and association of the fetal metabolome with newborn outcomes refer mostly to maternal overweight and obesity. The aim of this study was to investigate whether, as early as at the beginning of pregnancy, maternal pre-pregnancy underweight is associated with changes in the umbilical cord metabolome in a sample of the SNIP birth cohort [12].

## 2. Results

### 2.1. Baseline Characteristics of the Study Population

Table 1 shows the maternal and neonatal characteristics of the mother–child dyads (n = 239 with a ppBMI of <18.5 kg/m^2^ and n = 208 with a ppBMI of 18.5–24.9 kg/m^2^) in who the cord blood was analyzed. The maternal age was significantly different between the two groups, whereas the underweight mothers were at a median of 24 years old and the mothers of the control group were at a median of four years older. As expected, the ppBMI was significantly lower in the underweight group with 18.0 kg/m^2^ versus 21.5 kg/m^2^ in the normal weight group. Moreover, underweight mothers had a significantly lower placenta weight and smoked more than twice as often during pregnancy compared to the controls. The GWG showed no difference between the groups. As for the neonatal characteristics, the proportion of preterm neonates were higher in the group of mothers with a ppBMI of <18.5 kg/m^2^ (10% versus 0%), and the neonates showed significant differences in their birth weight.

### 2.2. Association of Maternal and Neonatal Parameters with Lipoprotein Subclasses

Figure 1 presents the association of maternal and neonatal parameters with the lipoprotein subclasses measured in neonatal cord blood.

Increasing ppBMI was associated with lower levels of LDL5 triglycerides, HDL4 cholesterol, LDL phospholipids, HDL phospholipids, HDL2 Apo-A1, HDL4 Apo-A1, and HDL4 Apo-A2, as well as higher levels of VLDL1 cholesterol. The associations for HDL4 cholesterol, HDL4 phospholipids, and HDL 2 and HDL 4 Apo-A1 stayed significant if the BMI was divided dichotomously at 18.5 kg/m^2^. Additionally, a ppBMI of <18.5 kg/m^2^ also showed a positive association with VLDL5 cholesterol but was negatively associated with VLDL triglycerides and HDL2 free cholesterol.

Increasing GWG resulted in higher levels of VLDL triglycerides and cholesterol, as well as phospholipids containing VLDL subclasses. Regarding HDL subclasses, only HDL1 Apo-A2 and HDL4 phospholipids were significantly associated with GWG.

When there was a ppBMI of <18.5 kg/m^2^ and a GWG below the 25th percentile combined, we observed decreased concentrations of total cholesterol along with decreased levels of the following cholesterol transporting lipoproteins: LDL4, LDL6, LDL free cholesterol, and HDL2 free cholesterol. LDL4 Apo-B, total Apo-A2, and HDL3 Apo-A2 were also decreased.

Seven lipoproteins were identified to have a negative association with the placenta-to-birth-weight ratio, especially the HDL subclasses transporting triglycerides, free cholesterol, and phospholipids that presented a negative ß-coefficient. A positive association with LDL2 cholesterol was detected for the placenta-to-birth-weight ratio.

Overall, no associations for total triglycerides, LDL triglycerides, HDL triglycerides, LDL cholesterol, and HDL cholesterol were found.

A subgroup analysis of the cord blood metabolome was performed in the group with a maternal ppBMI of <18.5 kg/m^2^. In this subgroup, the LDL1 Apo-B and LDL2 triglycerides were positively associated with ppBMI. The GWG showed a positive association with VLDL3 phospholipids.

Significant negative associations were observed for the placenta-to-birth-weight ratio and HDL trigylcerides, VLDL phopsholipids, and HDL phospholipids, as well as significant positive associations for LDL 2 cholesterol and LDL 3 cholesterol.

### 2.3. Association of Maternal and Neonatal Parameters with Plasma Metabolites

Figure 2 presents the association of maternal and neonatal parameters with plasma metabolites measured in neonatal cord blood. Decreasing continuous ppBMI is associated with decreasing values of tyrosine, valine, and lactic acid. Valine and lactic acid concentrations were also lower when ppBMI was dichotomized at 18.5 kg/m^2^. Increasing placenta-to-birth-weight ratios were associated with decreasing choline and alanine, as well as increasing formic acid. Increasing GWG was associated with increasing alanine, tyrosine, and lactic acid concentrations. GWG was negatively associated with 3-hydroxybutyric acid and acetone.

The group of ppBMI < 18.5 kg/m^2^ and GWG < 25th percentile combined showed lower concentrations of alanine and tyrosine.

A subgroup analysis of the cord blood metabolome was performed in the group with a maternal ppBMI of <18.5 kg/m^2^. Here, decreasing ppBMI was associated with decreasing valine and lactic acid.

In this subgroup, increasing the placenta-to-birth-weight ratio was associated with increasing formic acid. Increasing GWG was associated with increasing alanine and decreasing acetone.

## 3. Discussion

Our study focused on the cord blood metabolome of the newborns of mothers with pre-pregnancy underweight in a developed country. We were able to reveal significant associations between the early maternal variable ppBMI per se and cord blood metabolome, including lipoproteins, amino acids, and carboxylic acids. While this is in accordance with other investigations that mainly reported on the associations between pre-pregnancy overweight and obesity and the metabolome [30,31,32], most of the findings were related to HDL and LDL subclasses in our study. A low ppBMI was mainly significantly associated with higher concentrations of cord blood metabolites when compared to normal ppBMI, i.e., HDL phospholipids, LDL phospholipids, VLDL cholesterol, HDL cholesterol subclasses, and HDL 2 and HDL 4 Apo-A1. Unlike in adults, fetal HDL is the main lipoprotein class in cord blood [33,34]. One role of fetal HDL is to transport large amounts of cholesterol to the rapidly growing fetal tissues; therefore, it mainly takes up cholesterol from the placenta [35]. HDL can even influence the gene expression of the placenta tissue [35,36]. However, since the structure and function of fetal HDL differs from that of adult HDL, it is questionable how these early adaptations may affect later dyslipidemia. The second role of fetal HDL is to efflux the cholesterol from cells. Clinical studies have shown a correlation between cholesterol efflux capacity and the incidence and prevalence of cardiovascular disease, and it even appears to be independent of HDL-C concentration [37]. The cholesterol efflux capacity is also dependent on phospholipids. Investigations has shown that the concentration of HDL phospholipids is correlated to cholesterol efflux capacity [37]. Therefore, a possible explanation for higher HDL phospholipids in neonates of mothers with a low ppBMI might be the need for a greater capacity to distribute cholesterol. Studies have identified the association between phospholipids and insulin resistance and obesity [38]. Not only it is their concentration but their composition that has been described as important [39]. In a metabolome study, the composition of different phospholipids allowed researchers to distinguish between obese and normal-weight children between 6 and 15 years of life; thus, they may be considered as potential biomarkers of obesity [40].

GWG is another maternal factor, including obesity and metabolic syndrome, influencing the short and long-term outcomes of offspring [41]. The risk for developing obesity in the offspring of mothers with excessive GWG is not only elevated for adulthood, but also for childhood and adolescence [42]. In our study, significant associations between GWG and triglycerides, phospholipid, and cholesterol transporting VLDL were found. The transporting subclasses were positively associated with GWG. A possible explanation could be an abundance of lipids. Particularly, VLDL and HDL might be positively associated with GWG because these lipoproteins are necessary for transporting lipids and cholesterol to fetal tissues. Another possible explanation would be epigenetic modifications as a consequence of an altered maternal lipid status [43]. With additional risk factors, e.g., improper diet, lack of exercise, etc., epigenetic modifications that occurred during fetal life could contribute to dyslipidemia in adulthood [44,45,46].

The metabolic pattern changed when maternal pre-pregnancy underweight was combined with poor GWG (<25th percentile), which then was only significantly associated with decreased concentrations of metabolites, i.e., total cholesterol, LDL cholesterol, LDL and HDL free cholesterol, and several apoproteins. We assumed an aggravation in the deficiency of maternal lipids when both variables, i.e., a ppBMI of <18.5 kg/m^2^ and a poor GWG (<25th percentile), were combined, resulting in lower concentration of total cholesterol and its transporting lipoproteins. The adaptation mechanisms that could still compensate for a minor cholesterol deficiency (e.g., in mothers with a ppBMI of <18.5 kg/m^2^ and adequate GWG) are no longer effective, and there is a general reduction in cholesterol and its lipoproteins. These compensatory mechanisms may be driven by DNA methylation, which is influenced by maternal lipid status. However, the resulting stimulated gene expression can only compensate for the substrate deficiency in the fetal period to a limited extent. Kelishadi et al. [47] also found significantly lower concentration of total cholesterol in SGA neonates. In contrast, Miranda et al. [23] described higher plasma concentrations of cholesterol and the triglycerides transporting lipoproteins in IUGR. The authors speculated on an early link to cardiometabolic risk in later life due to an early dyslipidemic profile. In fetal circulation, the role and distribution of lipoproteins are different from that of adults. In contrast to adult metabolism, fetal HDL transports freshly synthesize the cholesterol from the liver to cells; additionally, HDL is the main fraction of the lipoproteins, whereas the LDL concentration is lower in the fetus than in adults [35,48]. Therefore, we consider that our results reflect an adapted metabolism to cholesterol deficit in the fetuses of women with a low GWG and low ppBMI. The substrate deficiency may have resulted in lowered LDL concentration since there was not enough cholesterol to transport. The liver might still be able to produce cholesterol, but this is transported by HDL to the tissues. Our subgroup analysis contradicts the results of previous studies that focused on fetal growth retardation and SGA with presenting higher concentrations of triglycerides, cholesterol, and LDL [23,49,50,51]. Our work used a ppBMI of <18.5 kg/m^2^ as an independent variable, whereas the other studies adjusted for a ppBMI when it was available and did not consider GWG. The causes for fetal growth restriction and SGA were not mentioned. Therefore, a different underlying mechanism may be assumed.

An increased, which means unfavorable, placenta-to-birth-weight ratio was shown to be related to an increased risk for coronary heart disease, and an abnormal placental size was associated with metabolic diseases in adulthood [52,53]. In our study, an increased placenta-to-birth-weight ratio also mainly resulted in decreased concentrations of seven metabolites (e.g., HDL phospholipids and LDL triglycerides, among others) but with a different pattern when compared to a low ppBMI combined with low GWG. Since an increase in the placenta-to-birth-weight ratio is an expression of a more insufficient working placenta, declining HDL phospholipids and LDL triglycerides might be a consequence of an impaired nutrient supply caused by the placenta. As discussed above, lower HDL phospholipids indicate a lower cholesterol efflux capacity, which might affect fetal growth. With respect to long-term outcomes, recent studies have suggested that placental shape and size influences the outcome with placental weight in relation to birth weight, and it can be used to predict later cardiovascular disease [53,54,55].

When repeating the analysis in the subgroup of newborns born to mothers with pre-pregnancy underweight only, an increasing placenta-to-birth-weight ratio was negatively associated with eleven metabolites of the lipidome (such as HDL triglycerides, HDL phospholipids, VLDL phospholipids, etc.), indicating even more of an imbalance in fetal supply, presumably as a consequence of low maternal lipid reserves. The Copenhagen Baby Heart Study reported that, at 14–16 months, children had reached adult concentrations for most lipid traits, and the children with high concentrations of atherogenic lipid parameters at birth had high concentrations at 2 months and 14–16 months [56].

In contrast to the metabolites of lipidome, several low weight molecules had to be excluded in our analysis due to their degradation when stored at −20 °C. For the remaining metabolites, we could reveal that maternal ppBMI was positively associated with tyrosine and valine. Tyrosine is not an essential amino acid, and its biosynthesis occurs from the essential amino acid phenylalanine. Tyrosine is necessary for the production of catecholamines, and current evidence shows that tyrosine is altered in patients with cardiovascular disease and/ or diabetes mellitus type 2 [57]. Valine, as part of the BCAAs, is an essential amino acid and cannot be synthesized by the human body itself. Apart from their physiological metabolism, BCAAs are associated with cardiometabolic diseases, neurodegeneration, insulin resistance, and diabetes mellitus [58,59,60]. However, it remains unclear whether BCAAs are actively involved in developing disease or if they are passive biomarkers. Interestingly, a review of Yao et al. [61] hypothesized that a lower concentration of valine results into fetal growth impairment. Since a lower birth weight is also related to a low ppBMI, a decreased level of valine could be a passive biomarker for impaired growth, or it could directly contribute to it. Continuos GWG was positive, and the combined categorial variable of ppBMI < 18.5 kg/m² and GWG < 25th percentile was negatively associated with tyrosine and alanine. Alanine functions as a precursor and regulator in fetal gluconeogenesis. A lack of alanine may increase the risk of a lower birth weight when considering its impact on the gluconeogenesis of a fast-growing fetus [62]. Choline status has been positively associated with weight and fat mass in human studies [63]. Accordingly, choline was negatively associated with the placenta-to-birth-weight ratio in our study.

The strength of our study is the large number of cord blood samples of newborns of mothers with a ppBMI of <18.5 kg/m^2^, and the detailed clinical phenotype of mothers and their newborns. In addition, this is the first study, to the authors’ knowledge, which analyzed the lipoprotein pattern in cord blood in such detail and focused on the influential variable of a maternal ppBMI of <18.5 kg/m^2^. Furthermore, the follow-up data enabled further analysis, which involved linking the cord blood metabolome to outcomes in childhood [64]. One limitation was the observational nature of this study, which complicates causal interpretations of the associations between maternal phenotypes, the newborn metabolome, and newborn outcomes. Due to the exploratory nature of our analysis, it is not suitable for an ad hoc power analysis. Hoenig et al. pointed out that “observed power can never fulfill the goals of its advocates because the observed significance level of a test also determines the observed power; for any test, the observed power is a 1:1 function of the *p* value” (Figure A1) [65]. In other words, as the *p*-value increases, the observed power decreases to 0. We have, therefore, refrained from conducting a post hoc analysis as this would not be helpful for the interpretation of our results. Another limitation was the storage condition of our blood samples with respect to time and temperature. This allowed only a subset of metabolites as opposed to a broader analysis of all metabolites. Finally, the findings await replication in other cohorts.

In conclusion, maternal underweight as early as at the beginning of pregnancy already influences the cord blood metabolome of the offspring. However, the metabolic pattern changes to the opposite when a pregnancy is complicated by a poor GWG or an unfavorable placenta-to-birth-weight ratio following pre-pregnancy underweight. Therefore, the pre-pregnancy maternal phenotype and, in turn, the different trajectories during fetal life are associated with different metabolic phenotypes of the newborn. This should be considered by future studies to improve neonatal outcomes after pregnancies are complicated by maternal underweight.

## 4. Materials and Methods

### 4.1. Study Design and Participants

A baseline recruitment of SNiP-I (“Survey of neonates in Pomerania”) was performed from March 2003 to November 2008. That study focused on collecting data regarding maternal and neonatal health, morbidity, and mortality. The details were reported by Ebner et al. [66]. A follow-up was performed at the age of 9–15 years [67].

The data of 5800 mother–child dyads were included in the baseline of SNiP-I. For this analysis, only participants with a recorded ppBMI and singleton pregnancies with life birth were included. Mothers who were overweight or obese (n = 1256) were excluded. Moreover, 13.9% (n = 475) of the remaining participants (n = 3401) had a ppBMI of <18.5 kg/m^2^. In another step, the mother–child dyads with missing smoking status and gestational weight gain were also excluded. From the remaining participants, cord blood was analyzed in 53.5% (n = 239) of cases. A sample of healthy mother–child dyads with a normal ppBMI (n = 208, a ppBMI of 19–24.9 kg/m^2^) and cord blood samples served as controls. Figure 3 gives information on the how mother–child dyads were excluded for analysis, and how the study population was separated into different groups.

### 4.2. Definition of Smoking

For this analysis, smoking behavior was categorized as “smoker/non-smoker”. Women smoking during the last four weeks before delivery were defined as “smoker”.

### 4.3. Definition of Small-for-Gestational Age (SGA) and Large-for-Gestational Age (LGA)

SGA was defined as a birth weight below the 10th percentile adjusted for gestational age; equally, LGA was defined as a birthweight above the 90th percentile [68].

### 4.4. Diagnosis of Neonatal Hypoglycemia

According to the German national guidelines, a plasma glucose concentration below 45 mg/mL (2.5 mmol/L) within the first 24 h postnatal was defined as hypoglycemia. Blood glucose was routinely measured in newborns with the following risk factors: preterm birth, low birth weight, SGA, LGA, and being born to mothers with diabetes in pregnancy.

### 4.5. Placental Weight/Birth Weight Ratio

Placental weight alone may be a crude proxy for its function. However, placental size is correlated with birth size. Therefore, the ratio between placental weight and birth weight has been shown as a useful indicator for placental efficiency [69]. A comparatively large placenta relative to birth weight may be an expression of a relatively inefficient placenta. Furthermore, the placenta-to-birth-weight ratio has been positively associated with cardiovascular disease mortality in adult life [70]. Therefore, the placenta-to-birth-weight ratio was calculated as the quotient of placental weight (g) divided by the birth weight (g).

### 4.6. Sample Collection and Measurement

The venous cord blood was collected by midwifes in EDTA tubes. The samples were centrifuged at 1900× *g* for 10 min to separate into plasma and cellular parts, which were then stored in 0.5 mL aliquots at −20 °C. Moreover, 250 µL of plasma from each aliquot was mixed with 250 µL of phosphate buffer solution. The buffer solution was prepared from deuterium, as well as sodium 3-trimethylsilyl-(2,2,3,3-D4)-1-propionate (TSP). The pH of the buffer solution was 7.4. For the 1H nuclear magnetic resonance (NMR) measurements, each sample was placed in a 5 mm NMR tube (Bruker Biospin, Rheinstetten, Germany). Measurements were recorded using a Bruker AVANCE-II 600 NMR spectrometer, which was operated by TOPSPIN 3.2 software and equipped with a 5 mm Z-gradient probe and an automatic tuning and matching assembly (ATMA). The sample tubes were automatically delivered to the spectrometer by Sample Jet. All the above components were supplied by Bruker Biospin, Rheinstetten, Germany. The acquisition temperature was set to 310 °K. A standardized one-dimensional pulse sequence with the following formula was used for measurement: -RD-gz,1-90°-t-90°-tm-gz,2-ACQ. Here, RD corresponds to the relaxation delay (4 s); t is a short delay (~3 µs); 90° represents the 90° RF pulse; tm is the mixing time (10 ms); gz,1 and gz,2 are the z-gradients of the magnetic field, which were both applied for 1 ms; and ACQ is the acquisition time (2.7 s). The water signal was suppressed using NOESYPRESAT, and a total of 98,304 measurement points were acquired with a total spectral width of 30 ppm. The receiver gain was set to 90.5 for all experiments. For pre-processing, a line broadening of 0.3 Hz, a zero fill to generate 128k data points, and a first-order phase correction of 0.0 were applied. The spectral processing included zero adjustment, line broadening, Fourier transform, chemical shift referencing, and a determination of spectral intensity per 1 mmol protons for quantitative referencing. The chemical shifts of the plasma spectra were referenced to the CH_3_ group signal of alanine and adjusted to 1.48 ppm. Finally, the spectrum was subjected to data analysis for lipoprotein subclass analysis B.I.LISATM (Bruker BioSpin GmbH, Rheinstetten, Germany), and the amino acids and sugars were analyzed B.I.Quant-PSTM (Bruker BioSpin GmbH, Rheinstetten, Germany).

After preprocessing, data from 450 subjects were recorded, including 38 metabolites and 112 measurements of lipoproteins covering the entire density gradient from very low-density particles (VLDL; 0.950–1.006 kg/L), low-density particles (LDL, density 1.09–1.63 kg/L), to high-density particles (HDL; density 1.063–1.210 kg/L). The lipoprotein main fractions were further subdivided into different density subclasses (see Appendix B). Along with their distinction in density subclasses, important chemical components such as cholesterol, triglyceride, phospholipid, and free cholesterol content, as well as measurements of major apolipoproteins, were included and used for statistical analysis. Furthermore, 18 out of the 38 metabolites had a detection rate of <25% and were excluded from the analyses (Table A1).

### 4.7. Impact of Storage Temperature on NMR Metabolomics

In SNiP, plasma samples were stored at −20 °C. Long-term storage at −20 °C limits the analysis of several metabolites [71]. To identify metabolites that are affected by storage temperature, two comparison studies were performed in advance: First, fresh cord blood samples were aliquoted and stored over one week at −20 °C, as well as at −80 °C. Second, each of the healthy term neonates in the 80 samples of SNiP (−20 °C, collected 2002–2008) and 80 samples from a second cohort (SNiP-II, stored at −80 °C, collected 2013–2017) were matched for maturity, weight, and sex. For both studies, the metabolites and lipoproteins measured by 1H NMR were compared with respect to the different conditions. The analyses revealed that the lipoproteins were stable over time at different temperatures. However, some of the low weight metabolites needed to be excluded from the following analyses due to degradation (Table A1).

For the investigated 20 metabolites, we had stability data for 17 metabolites, with 9 metabolites being stable at −20 °C. In total, we included 12 metabolites in the subsequent analyses.

### 4.8. Statistical Analysis

The continuous data were expressed as the median (Q1; Q3). The nominal data were displayed as total numbers and percentages. For the bivariate analyses, the Kruskal–Wallis test (continuous data) or χ^2^-test (nominal data) were performed. Before checking for associations, we performed Pearson correlation tests for the measured lipoprotein subclasses (Figure 4). Based on the correlations, some parameters were not considered further to minimize the statistical error rate. Linear regression models were used to assess the associations between the cord blood metabolites/lipoproteins (dependent variables) and the independent variables, including a ppBMI of <18.5 kg/m^2^, a continuous ppBMI, GWG, and the placental ratio (the placenta weight divided by birth weight). These were then adjusted for the maternal parameters of age, sex of child, smoking during pregnancy, gestational age, and neonatal parameters (birth weight and hypoglycemia), which were used to detect the associations between. Subgroup analyses using the same linear regression models were performed among the mother–child dyads with a maternal ppBMI of <18.5 kg/m^2^ and a GWG < 25th percentile. Furthermore, the analyses were repeated in the subgroup of women with a ppBMI of <18.5 kg/m^2^. A *p*-value below 0.05 was considered statistically significant. In the association studies, to account for multiple testing, we further adjusted the *p*-values from regression analyses by controlling the false discovery rate (FDR) at 5% using the Benjamini–Hochberg procedure. Statistical analyses were performed with SAS 9.4 (SAS Institute Inc., Cary, NC, USA).

## Figures and Tables

**Figure 1 ijms-25-07552-f001:**
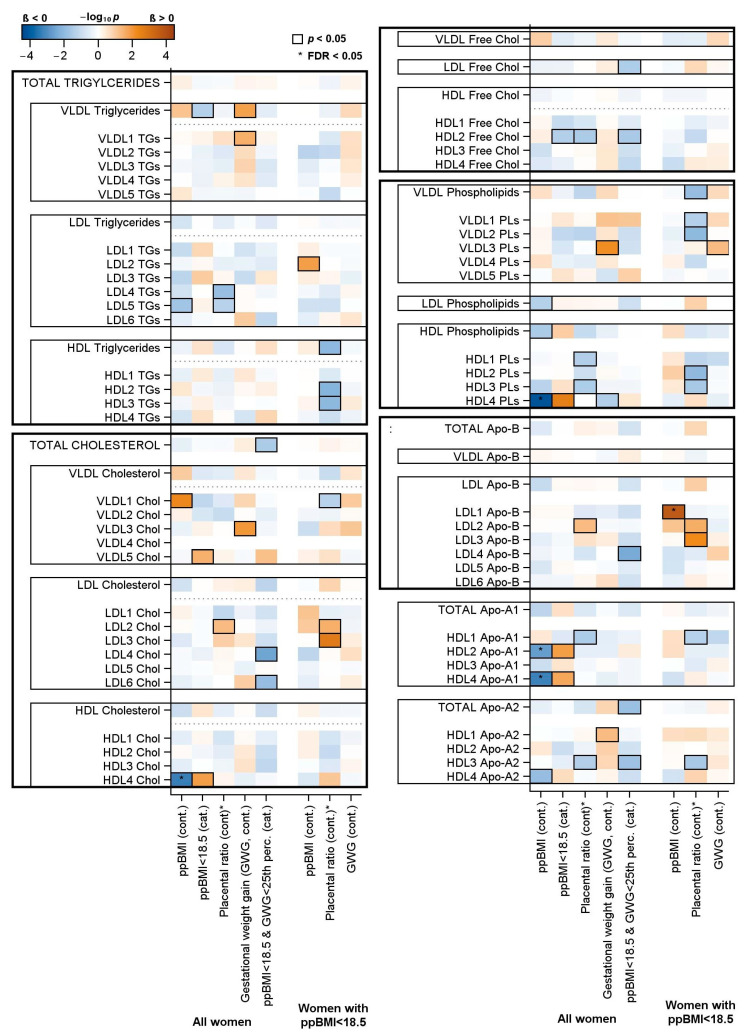
Color-coded *p*-values for linear regression models of the association of maternal and neonatal parameters with lipoprotein subclasses. The parameters include the following: (1) pre-pregnancy BMI (ppBMI) as a continuous, as well as categorized, variable (a ppBMI of <18.5 kg/m^2^); (2) the placenta-to-birth-weight ratio (placental ratio); (3) the gestational weight gain as a continuous variable; and (4) the combination of a low ppBMI (<18.5 kg/m^2^) and a low gestational weight gain (<25th percentile) as a categorized variable. The analyses were partly repeated in the subgroup of women with a ppBMI of <18.5 kg/m^2^. Significant associations (*p* < 0.05) are marked with a black box. Associations that were significant after controlling the false discovery rate (FDR) (FDR < 0.05) are marked with *. The models were adjusted for age, sex of child, smoking during pregnancy, gestational age, birth weight, neonatal hypoglycemia, and additionally for the ppBMI in models for placental ratio and gestational weight gain.

**Figure 2 ijms-25-07552-f002:**
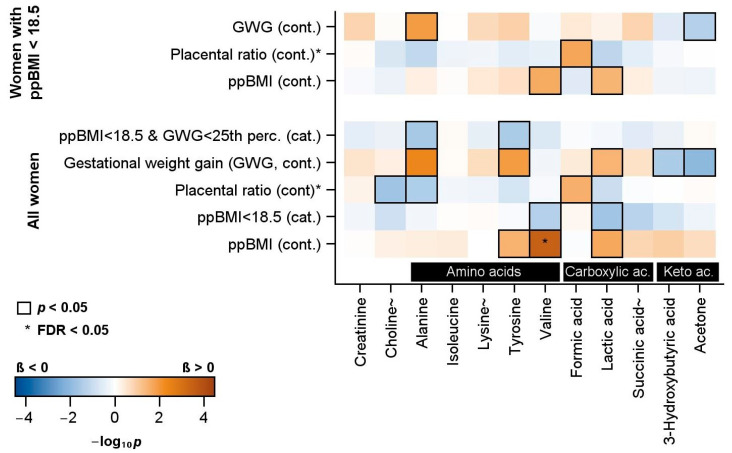
Color-coded *p*-values for the linear regression models of the association of maternal and neonatal parameters with plasma metabolites. The parameters included the following: (1) pre-pregnancy BMI (ppBMI) as a continuous, as well as categorized, variable (a ppBMI of <18.5 kg/m^2^); (2) the placenta-to-birth-weight ratio (placental ratio); (3) the gestational weight gain as a continuous variable, and (4) the combination of a low ppBMI (<18.5 kg/m^2^) and a low gestational weight gain (<25th percentile) as a categorized variable. The analyses were partially repeated in the subgroup of women with a ppBMI of <18.5 kg/m^2^. Significant associations (*p* < 0.05) are marked with a black box. Associations that were significant after controlling the false discovery rate (FDR) (FDR < 0.05) are marked with *. The models were adjusted for adjusted for age, sex of child, smoking during pregnancy, gestational age, birth weight, neonatal hypoglycemia, and additionally for ppBMI in the models for placental ratio and gestational weight gain. ~ Metabolites without stability data.

**Figure 3 ijms-25-07552-f003:**
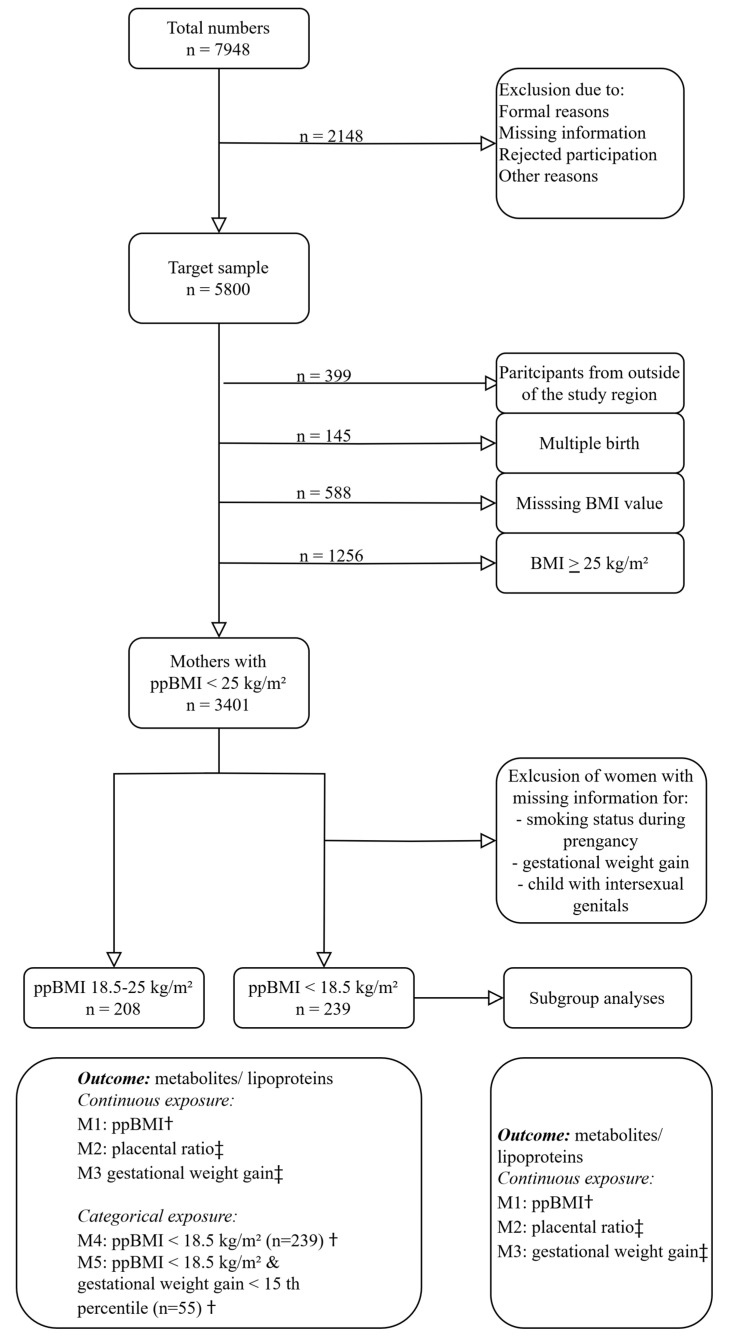
Flow diagram showing the selection process and statistical analysis for the analyzed sample of the SNiP birth cohort. † adjusted for age, sex of child, smoking during pregnancy, gestation age, birth weight, and neonatal hypoglycemia. ‡ adjusted for age, sex of child, smoking during pregnancy, gestation age, birth weight, and neonatal hypoglycemia.

**Figure 4 ijms-25-07552-f004:**
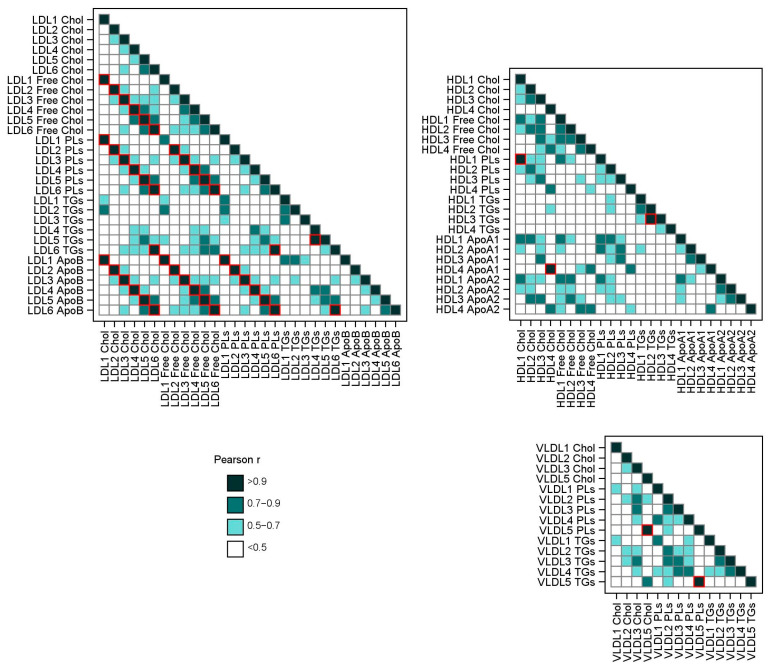
Correlation of lipoproteins using Pearson correlation coefficient r.

**Table 1 ijms-25-07552-t001:** SNiP birth cohort. The maternal and neonatal characteristics of women with pre-pregnancy underweight and normal weight.

		Pre-Pregnancy BMI			
	n	Underweight (BMI < 18.5 kg/m^2^)	n	Normal Weight (BMI 18.5–25 kg/m^2^)	*p* *
Maternal age, years	239	24 (21; 28)	208	28 (24; 32)	<0.01
Female sex of child, n (%)	239	106 (44.4)	208	94 (45.2)	0.92
Pre-pregnancy BMI, kg/m^2^	239	18.0 (17.4; 18.3)	208	21.5 (20.1; 22.7)	<0.01
Placenta weight, g	125	520 (450; 590)	103	555 (485; 610)	0.01
Smoking during pregnancy, n (%)	239	94 (39.3)	208	33 (15.9)	<0.01
Neonatal hypoglycemia, n (%)	239	7 (2.9)	208	2 (1.0)	0.18
Gestational weight gain, kg	239	16.0 (13.0; 19.0)	208	16.0 (13.0; 20.0)	0.32
Gestational age, n (%)	239		208		<0.01
<32 weeks 32–36 weeks 37–41 weeks >41 weeks		6 (2.5) 18 (7.5) 212 (88.3) 4 (1.7)		0 0 201 (96.6) 7 (3.4)	
Birth weight, n (%)	239		208		< 0.01
AGA SGA LGA		195 (81.3) 34 (14.2) 11 (4.6)		204 (98.1) 2 (1.0) 2 (1.0)	

Data are expressed as the median, 25th, or 75th percentile (continuous data), as well as absolute numbers and percentages (categorical data). AGA: appropriate-for-gestational-age. SGA: small-for-gestational-age. LGA: large-for-gestational-age. * Chi-quadrat tests (nominal data) or Kruskal–Wallis tests (interval data) were performed.

## Data Availability

This paper is based on the data collected during the study ‘Survey of Neonates in Pomerania’, which was conducted at the University Medicine Greifswald, Greifswald, Germany, between 2002 and 2008. The repository is managed by the Research Cooperation Community Medicine (RCC) of the University of Greifswald, Germany. This data repository allows any researcher to register and apply for access. It provides a data dictionary and online application tools for accessing the data. Upon application by the registered users, the RCC determines whether to grant access to the data, based on scientific guidelines.

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
