# Peer review of "Impact of Maternal Pre-Pregnancy Underweight on Cord Blood Metabolome: An Analysis of the Population-Based Survey of Neonates in Pomerania (SNiP)"

_ijms, 2024, doi:10.3390/ijms25147552_

Round 1

Reviewer 1 Report

[review text omitted: it was posted to a different submission]

Author Response

Congratulations on your work!

Response: We thank the reviewer for the positive reception of our revised manuscript.

Comment: The title should state that this is a review of the current literature.

Response: It was not our aim to publish a review. Although we used quite a lot of references for our work, we wrote this paper with the goal of publishing an original article. Moreover, the work does not fulfill the scope of a review.

Comment: The introduction is clear and detailed, but I think it would be useful to state the aim of the article.

Response: Thank you for your comment. The aim of our work was to investigate the impact of maternal pre-pregnancy underweight to the cord blood metabolome of the newborn reflecting its fetal metabolism. The statement was made at the end of the introduction (line 100 – 103):

‘ […] The aim of this study was to investigate whether as early as at the beginning of pregnancy, maternal pre-pregnancy underweight is associated with changes in the umbilical cord metabolome in a sample of the SNIP birth cohort [12]. ‘

Comment: The other chapters are well elaborated. I consider that more images from ultrasound examination or MRI would be helpful. Perhaps replacing or corroborating written information with photos would be a better idea. Structuring the article in chapters is a good idea, but a discussions one would be useful, giving a perspective of all the data stated in the article.

Response: We did not consider including pictures of MRI or ultrasound because it would not match our work since we used metabolome analyses of different metabolites and lipoproteins.
We would like to point out that there has probably been a mix-up in the allocation. We have the feeling that the reviewer's comments are not related to our work. If we are mistaken in this regard, please accept our apologies and we hope that our answers are satisfactory.

Reviewer 2 Report

Comments and Suggestions for Authors

The manuscript is well-written and organized. However, there are some issues. After addressing those issues, the manuscript can be published. 

Abstract:OK

Introduction: The fetal metabolome notion should be expended in a few phrases

Results: OK

Discussions: Discussions regarding metabolic patterns should be expanded in a few phrases.

                       The study's strength should be further emphasized in a few phrases to increase the value of the manuscript.

 Materials and methods: The flow chart (Figure 3) must be redrawn to produce a complete flow, easy-to-read methodology chart. Also, inclusion and exclusion criteria must be added.

                                              The power of the study must be computed.

                                               Some specifications about the statistical significance of the study must be added here or in the discussion sections

Author Response

The manuscript is well-written and organized. However, there are some issues. After addressing those issues, the manuscript can be published.

Response: We thank the reviewer for the positive reception of our revised manuscript.

Introduction:

Comment: The fetal metabolome notion should be expended in a few phrases.

Response: We thank the reviewer for the opportunity to elaborate on the importance of the fetal metabolome. We have therefore described in more detail the important interface that the analysis of the fetal metabolome occupies in current research (line 74 – 77).

‘[…] The fetal metabolome comprises the totality of all metabolites at a specific point in time. Analysis of the fetal metabolome provides the opportunity to reconcile fetal gene expression, epigenetic changes and environmental influences with the metabolic phenotype of the fetus, and new insights into the fetal metabolome therefore enable a better understanding of physiological and pathological processes. […]’

Discussion:

Comment: Discussions regarding metabolic patterns should be expanded in a few phrases.

Response: Although we are able to provide a more detailed description of the biological metabolic patterns with regard to lipoproteins, the rather limited data available for this detailed lipoprotein breakdown makes it difficult to draw concrete conclusions. A more precise breakdown of the metabolic pathways requires further work steps that are not possible with our data set. Nevertheless, there are indications in the literature that, for example, the lipidome of the mother causes epigenetic changes in the fetus. This might result in adaptation mechanisms due to enhanced gene expression. Our results possibly display the metabolic phenotype of these epigenetic processes or even its limits, which is why we have added the following section (line 239 – 247):

‘[…] We assume an aggravation in deficiency of maternal lipids when both variables, ppBMI <18.5 kg/m² and poor GWG (<25th percentile), are combined resulting in lower concentration of total cholesterol and its transporting lipoproteins. Adaptation mechanisms that could still compensate for a minor cholesterol deficiency (e.g. in mothers with ppBMI < 18.5 kg/m² and adequate GWG) are no longer effective and there is a general reduction in cholesterol and its lipoproteins. These compensatory mechanisms may be driven by DNA methylation, which is influenced by maternal lipid status [52]. However, the resulting stimulated gene expression can only compensate for the substrate deficiency in the fetal period to a limited extent.[…]’

Comment: The study's strength should be further emphasized in a few phrases to increase the value of the manuscript.

Response: Done. (line 310-312)

‘ […] In addition, this is the first study to the authors knowledge, which analyzed the lipoprotein pattern in cord blood in such detail and focusing on the influential variable of maternal ppBMI < 18.5 kg/m². […]’

Materials and methods:

Comment: The flow chart (Figure 3) must be redrawn to produce a complete flow, easy-to-read methodology chart. Also, inclusion and exclusion criteria must be added.

Response: As requested, we have redrawn Figure 3 now with detailed description of the selection process of our study group with additional information of statistical analysis.

Figure 3. Flow diagram showing the selection process and statistical analysis for the analyzed sample of the SNiP birth cohort. † adjusted for age, sex of child, smoking during pregnancy, gestation age, birth weight, neonatal hypoglycemia. ‡ adjusted for age, sex of child, smoking during pregnancy, gestation age, birth weight, neonatal hypoglycemia.

Comment: The power of the study must be computed.

Response: We agree with the reviewer that a power analysis could provide important information for future studies. However, we would like to highlight that the exploratory nature of our analysis  is simply not suitable for an ad-hoc power analysis. The underlying conceptual problem is that: “Observed power can never fulfill the goals of its advocates because the observed significance level of a test also determines the observed power; for any test, the observed power is a 1:1 function of the P value.” [1]. We would like to highlight that tests with small P-values always have high “observed” power. As the P-value increases, the “observed” power decreases to 0. If one considers post hoc power an essential adjunct to the P-value, most studies will be interpreted as either:

  1. “We saw evidence of a treatment effect, and it’s a strong finding because our study has high power” (because studies with small P-values will always have high “observed” power), or
  2. “We did not see evidence of a treatment effect, but it was probably because our study has low power” (because studies with large P-values will always have low “observed” power).”

Hence, we feel that the suggestion to perform a post-hoc power analysis is not just inappropriate but would also not be very helpful with regards to interpreting our results.

Citation:
[1] Hoenig, John M., and Dennis M. Heisey. "The abuse of power: the pervasive fallacy of power calculations for data analysis." The American Statistician 55.1 (2001): 19-24.

Comment: Some specifications about the statistical significance of the study must be added here or in the discussion sections

Response: We agree to specify the statistical significance of our findings and added it in the discussion section.

Round 2

Reviewer 2 Report

Comments and Suggestions for Authors

The manuscript has been improved. The response regarding the study's power should be included in the discussion section. 

Author Response

Reviewer 2

Comment: The manuscript has been improved. The response regarding the study's power should be included in the discussion section.

Response: We are pleased that our corrections are satisfactory. As you requested, we implemented the response regarding the study’s power in a few phrases (line 319 - 323):

 […] Hoenig et al. pointed out that “observed power can never fulfill the goals of its advocates because the observed significance level of a test also determines the observed power; for any test, the observed power is a 1:1 function of the P value” (Figure 1A) [65]. In other words, as the P-value increases, the observed power decreases to 0. We have therefore refrained from conducting a post-hoc analysis, as this would not be helpful for the interpretation of our results. […]